# Prognostic Value of Neutrophil-To-Lymphocyte Ratio and Platelet-To-Lymphocyte Ratio for Renal Outcomes in Patients with Rapidly Progressive Glomerulonephritis

**DOI:** 10.3390/jcm9041128

**Published:** 2020-04-15

**Authors:** Yukari Mae, Tomoaki Takata, Ayami Ida, Masaya Ogawa, Sosuke Taniguchi, Marie Yamamoto, Takuji Iyama, Satoko Fukuda, Hajime Isomoto

**Affiliations:** Division of Medicine and Clinical Science, Faculty of Medicine, Tottori University, Yonago, Tottori 683-8504, Japan

**Keywords:** NLR, PLR, RPGN, predictive value, hemodialysis, withdrawal, cellular crescent, global sclerosis

## Abstract

Background: Rapidly progressive glomerulonephritis (RPGN) is a syndrome characterized by a rapid decline in renal function that often causes end-stage renal disease. Although it is important to predict renal outcome in RPGN before initiating immunosuppressive therapies, no simple prognostic indicator has been reported. The aim of this study was to investigate the associations of neutrophil-to-lymphocyte ratio (NLR) and platelet-to-lymphocyte ratio (PLR) to renal outcomes in patients with RPGN. Methods: Forty-four patients with a clinical diagnosis of RPGN who underwent renal biopsy were enrolled. The relationships between NLR and PLR and renal outcome after 1 year were investigated. Results: NLR and PLR were significantly higher in patients with preserved renal function in comparison to patients who required maintenance hemodialysis (*p* < 0.05 and *p* < 0.01, respectively). An NLR of 4.0 and a PLR of 137.7 were the cutoff values for renal outcome (area under the curve, 0.782 and 0.819; sensitivity, 78.4% and 89.2%; specificity, 71.4% and 71.4%, respectively). Furthermore, an NLR of 5.0 could predict recovery from renal injury in patients requiring hemodialysis (area under the curve, 0.929; sensitivity, 83.3%; specificity, 85.7%). Conclusion: NLR and PLR could be candidates for predicting renal outcomes in patients with RPGN.

## 1. Introduction

Rapidly progressive glomerulonephritis (RPGN) is a syndrome characterized by hematuria, proteinuria, anemia, and a rapid decline in renal function [1]. The diagnosis of RPGN is made when renal dysfunction occurs within a short period of time and is complicated with proteinuria or hematuria [2]. The etiology of RPGN is divided into three classifications: immune complex crescentic glomerulonephritis, pauci-immune crescentic glomerulonephritis, and anti-glomerular basement membrane (GBM) crescentic glomerulonephritis. In Japan, the number of end-stage renal disease (ESRD) cases caused by RPGN has increased approximately 3.1 times between 1994 and 2018, which represents the fifth most common etiology of ESRD [2,3]. Since RPGN causes a progressive decline in renal function, patients with RPGN require aggressive treatment with steroids and immunosuppressive agents [4]. However, these treatments are not always effective and, in such cases, RPGN is refractory and requires maintenance hemodialysis (HD). Considering that steroids and immunosuppressive agents can cause life-threatening infections, conservative treatment is also considered for patients with RPGN. Although it is critically important to predict renal outcomes in the early stages of RPGN [5], a simple prognostic marker for RPGN is yet to be established.

In recent years, neutrophil-to-lymphocyte ratio (NLR) and platelet-to-lymphocyte ratio (PLR) have received attention as potential new markers of systemic inflammation. In previous studies, NLR and PLR have been reported to be useful in systemic inflammatory diseases such as aortitis syndrome [6], Behçet’s disease [7], Kawasaki disease [8], Henoch–Schönlein purpura [9], systemic lupus erythematosus [10], and anti-neutrophil cytoplasmic antibody (ANCA)-associated vasculitis (AAV) [11,12]. Furthermore, NLR and PLR have been proposed as markers of inflammation in patients with ESRD [13,14]. Therefore, we speculated that NLR and PLR could be simple predictors of renal decline in RPGN. The purpose of this study was to investigate the associations of NLR and PLR to renal outcome in patients with RPGN.

## 2. Materials and Methods

### 2.1. Study Population

In this study, we enrolled 501 patients who underwent renal biopsy at the Tottori University Hospital between 2009 and 2019. Renal biopsies were performed according to the indications of the guidelines from the Japanese Society of Nephrology [15]; persistent hematuria and/or proteinuria, proteinuria more than 0.5 g/day, a rapid decline in renal function, or gross hematuria. Among the 501 patients enrolled, 47 patients were clinically diagnosed with RPGN based on the guidelines from the Japanese Society of Nephrology [16]. Excluding 2 cases with an active bacterial infection and 1 case with a relapse of the glomerulonephritis, 44 patients were included in the analyses (Figure 1). None of the patients included had a history of cancer or prescribed corticosteroids. Immunosuppressive therapies were determined according to the guidelines [16]. This study was conducted in accordance with the Declaration of Helsinki and approved by the Ethics Committee of Tottori University Hospital (approval number: 19A138).

### 2.2. Clinical and Laboratory Findings

The patient’s characteristics and laboratory findings on admission, including white blood cell count (WBC), neutrophil count (Neu), lymphocyte count (Lym), platelet count (Plt), creatinine (Cr), estimated glomerular filtration rate (eGFR) [17], C-reactive protein (CRP), erythrocyte sedimentation rate (ESR), myeloperoxidase (MPO)-ANCA, proteinase 3 (PR3)-ANCA, and the anti-GBM antibody, were acquired retrospectively. NLR was calculated as the ratio of neutrophil count to lymphocyte count (NLR = Neu/Lym), and PLR was calculated as the ratio of platelet count to lymphocyte count (PLR = Plt/Lym). Renal outcomes 1 year from diagnosis were also recorded.

### 2.3. Histological Findings

Ultrasound-guided renal biopsy was performed as previously described [18]. In brief, renal tissue was obtained using a 16-gauge biopsy gun (Acecut; TSK Laboratory, Tochigi, Japan). The specimen was fixed in 10% formalin and embedded in paraffin. Sections (4 µm thickness) were stained with periodic acid-Schiff (PAS). Pathological changes in glomeruli were defined as global sclerosis, cellular crescent, fibrocellular crescent, fibrous crescent, and others. Pathological analyses were performed by an experienced nephrologist (S.F.), who was independent of the acquisition and analysis of the clinical information.

### 2.4. Statistical Analyses

Continuous variables were expressed as the mean ± standard deviation or the median (range) according to the distribution. The Kolmogorov–Smirnov test was used to assess normal distribution. Differences between groups were analyzed using the Student’s t test for normally distributed variables, the Mann–Whitney U test for non-normally distributed variables, or the chi-square test for categorical variables. In addition, receiver operating characteristic (ROC) curve analysis was performed to determine the optimal cutoff values for NLR and PLR. The optimal cutoff point was determined by minimizing the square of the distance between the point (sensitivity of 1, 1-specificity of 0) and any point on the ROC curve. Multivariate regression analysis was carried out, in which age, eGFR, CRP, and NLR or PLR were selected, with the stepwise forward selection method, to investigate independent predictors of renal outcomes in the 44 patients. StatFlex Ver7 for Windows (Artec, Osaka, Japan) was used for the statistical analyses. A two-tailed *p*-value of < 0.05 was considered statistically significant.

## 3. Results

### 3.1. Differences between Patients with Preserved Renal Function and Renal Failure

All patients enrolled in this study were ethnically homogenous. The etiology of the 44 patients was as follows: ANCA-associated vasculitis (*n* = 34), ANCA-negative vasculitis (*n* = 6), and anti-GBM disease (*n* = 4). We first divided the patients into two groups according to their renal outcomes at 1 year post diagnosis. The characteristics of the 37 cases with preserved renal function (pre-dialysis group) and 7 cases with renal failure (maintenance HD group) are shown in Table 1 and Figure 2. WBC, Neu, Plt, Cr, eGFR and the anti-GBM antibody all showed significant differences between the groups. We also observed significant differences in NLR (8.2 (2.0–32.0) vs. 3.9 (2.8–8.4), *p* = 0.019) and PLR (265.7 (82.9–2255.0) vs. 126.0 (107.1–269.0), *p* = 0.008) between the pre-dialysis and maintenance HD groups, respectively. Multivariate regression analysis revealed that renal function was the strongest influencing factor for renal outcome (stdβ = 0.363, *p* = 0.012). There was also a trend suggesting the significance of NLR as a predictive value (stdβ = 0.276, *p* = 0.052); PLR, however, did not display this significance (stdβ = 0.207, *p* = 0.148).

The ROC curves analyses were performed to define the cutoff value of PLR and NLR for predicting renal outcomes after 1 year (Figure 3). Both NLR and PLR were accurate predictors of renal outcomes, with an area under the curve (AUC) of 0.782 in NLR and 0.819 in PLR. The cutoff values defined were 4.0 in NLR, with a sensitivity of 78.4% and specificity of 71.4%, and 137.7 in PLR, with a sensitivity of 89.2% and specificity of 71.4%.

### 3.2. Differences between Patients with Temporary Hemodialysis and Maintenance Hemodialysis

Since renal function on admission was a strong predicting factor for renal outcome, we divided the 13 patients who required HD into two groups as follows: 6 patients with recovery of renal function (temporary HD group) and 7 patients with persistent renal failure (maintenance HD group). Sex, WBC and Neu showed significant differences between the groups (Table 2). NLR was significantly higher in the temporary HD group compared to the maintenance HD group (12.4 (4.1–21.4) vs. 3.9 (2.8–8.4), *p* = 0.008, respectively, Figure 4). However, no significant difference was observed in PLR between the temporary HD group and the maintenance HD group (341.7 ± 217.7 vs. 156.1 ± 62.6, *p* = 0.053, respectively). The ROC curve analysis showed that an NLR of 5.0 could predict withdrawal from HD with a sensitivity of 83.3% and a specificity of 85.7%, with an AUC of 0.929 (Figure 5).

We further investigated histological changes in the temporary HD group and maintenance HD group (Table 3, Figure 6 and Figure 7). The number of globally sclerotic glomeruli was significantly lower in the temporary HD group (9.0% ± 10.1% vs. 53.0% ± 9.7%, *p* < 0.001), whereas the number of glomeruli with cellular crescent was significantly higher in the temporary HD group (27.9 (0–73.3) vs. 0 (0–13.3), *p* = 0.022).

## 4. Discussion

In this study, we found that NLR and PLR at the point of diagnosis of RPGN are associated with renal outcome. In particular, NLR was considered to be a useful prognostic indicator for the recovery from HD in patients with RPGN.

RPGN often causes a progressive decline in renal function that leads to ESRD at a high rate. In this study, we observed that around 16% of the cases resulted in ESRD. Several renal prognostic indicators of RPGN, such as the degree of decline in renal function on admission, histological classification, and the level of the anti-GBM antibody, have been suggested in previous reports [4,19]. However, it is difficult to accurately predict renal outcome without a renal biopsy or in patients who require HD. Therefore, it is important to establish a simple renal prognostic indicator other than renal function or histological assessment.

NLR and PLR are simple and cost-effective markers, that represent the ratio of the number of cells with two different hemocytes. Neu and Plt increase with inflammation [11,20], while Lym may decrease with inflammation in autoimmune diseases [21]. Since the majority of the patients included in this study had an etiology of autoimmune vasculitis, it was expected that the increase in Neu and Plt, and the decrease in Lym, would be proportionate to the degree of inflammation. Therefore, we considered that NLR and PLR could be more reliable than a single hemocyte number. Infection, cancers, ischemic heart disease and peripheral vascular disease affect NLR and PLR [22]. In addition, steroids increase Neu, while immunosuppressive agents may reduce Neu by myelosuppression. Thus, in this study, we excluded patients who had infectious diseases and who were already administered steroids or immunosuppressive drugs at diagnosis, and confirmed no patient had a history of malignancy, ischemic heart disease or peripheral vascular disease.

NLR and PLR have been reported to be associated with AAV disease activity; high NLR and PLR indicate a higher disease activity [11,12,20,22]. On the other hand, several studies have mentioned that the application of NLR and PLR is limited. It has been demonstrated that NLR is a good predictor of the relapse rate, but not of death in patients with AAV [22]. PLR is also able to predict the disease activity but cannot predict relapse in AAV patients [20]. In this study, both NLR and PLR at diagnosis were significantly higher in patients with preserved renal function than in patients with maintenance HD. We speculate that a higher NLR and PLR indicate acute disease and an active phase, sustaining the possibility of a positive response to immunosuppressive therapy, whereas a lower NLR and PLR may suggest a chronic phase with irreversible renal injury. This was confirmed by the histological analysis, which revealed significant differences in glomerular changes. The majority of the glomeruli in the maintenance HD group were globally sclerosing, indicating irreversible renal injury. Cellular crescent presence, suggesting a possibility of improvement, was highly observed in the temporary HD group. We demonstrated that an NLR < 4.0 or PLR < 137.7 at diagnosis were associated with negative renal outcomes, especially in patients requiring HD. An NLR < 5.0 at diagnosis could predict irreversible renal failure.

Since the patients in the pre-dialysis group showed variable renal function, and the multivariate analysis revealed that renal function was the strongest influencing factor, we investigated the predictive abilities of NLR and PLR in patients requiring HD. Among the 13 patients, NLR at diagnosis was significantly higher in the temporary HD group than in the maintenance HD group. Although PLR showed an increased presence in the temporary HD group, the difference was not significant. The half-life of Neu and Plt could affect this result. Neu can survive for less than 24 h, while Plt survives for 10 days, and their lifespans are controlled by endogenous apoptosis [23,24]. Plt, which is increased by inflammation, circulates for a longer period than Neu. In predicting the course of patients requiring HD, it would be desirable to evaluate the acute phase of inflammation and disease activity. Therefore, NLR would be a better predictor than PLR for withdrawal of HD.

There are some limitations to our study. First, all the patients were treated based on the clinical guidelines for the ANCA-associated RPGN [25]; thus, the treatment strategy differed in each patient. Since all four patients with an anti-GBM disease required maintenance HD, this may affect the result of our study. However, we observed a significant difference in NLR between the temporary and maintenance HD groups when these patients were eliminated. In addition to the variations in NLR and PLR, this study was a retrospective study, with a small number of subjects. Therefore, the results of the present study should be carefully interpreted, and a prospective study with a larger number of patients is required to confirm the suitability of NLR and PLR as predicative factors in renal outcomes.

## 5. Conclusions

In conclusion, we revealed that the NLR and PLR at diagnosis could predict renal outcomes in patients with RPGN, and that NLR could predict withdrawal from HD in patients requiring HD. Treatment strategies could be modified according to the NLR and PLR, especially in patients whose renal function is unlikely to recover, which may reduce the risk of treatment-related complications.

## Figures and Tables

**Figure 1 jcm-09-01128-f001:**
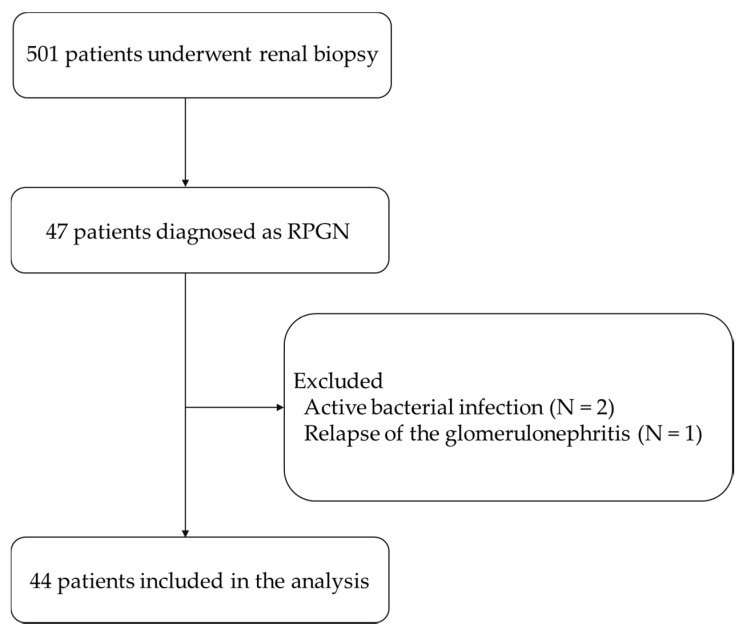
Study design. Of the 501 patients who underwent renal biopsy, 44 patients were included in the analysis.

**Figure 2 jcm-09-01128-f002:**
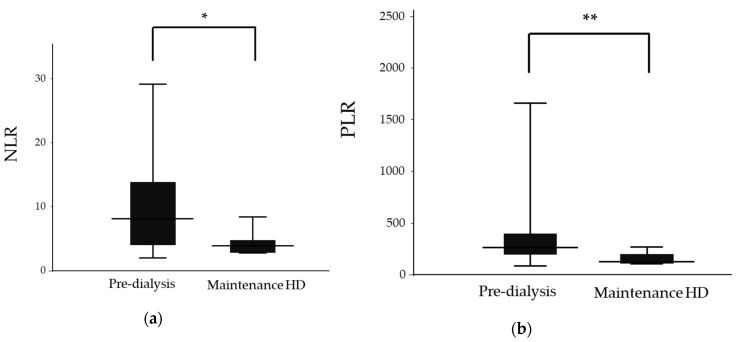
Neutrophil-to-lymphocyte ratios (NLR) and platelet-to-lymphocyte ratios (PLR) in the pre-dialysis and maintenance hemodialysis (HD) groups. (**a**) NLR in the pre-dialysis and maintenance HD groups. (**b**) PLR in the pre-dialysis and maintenance HD groups. The top and the bottom of the boxes are the first and third quartile, respectively. The length of the box represents the interquartile range. The line through the middle of each box represents the median. The error bars show the minimum and maximum values (range). *, *p* < 0.05; ** *p* < 0.01. NLR—neutrophil-to-lymphocyte ratio; PLR—platelet-to-lymphocyte ratio; HD—hemodialysis.

**Figure 3 jcm-09-01128-f003:**
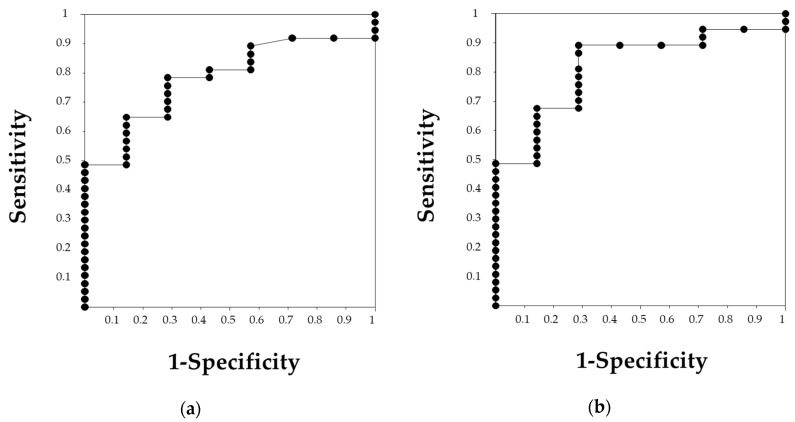
The receiver operating characteristic (ROC) curve of neutrophil-to-lymphocyte ratio (NLR) and platelet-to-lymphocyte ratio (PLR) for predicting renal outcome. (**a**) The ROC curve of NLR showing an area under the curve (AUC) of 0.782. An NLR of 4.0 was the cutoff value with a sensitivity of 78.4% and a specificity of 71.4%. (**b**) The ROC curve of PLR showing an AUC of 0.819. The cutoff value of 137.7 was determined with a sensitivity of 89.2% and a specificity of 71.4%.

**Figure 4 jcm-09-01128-f004:**
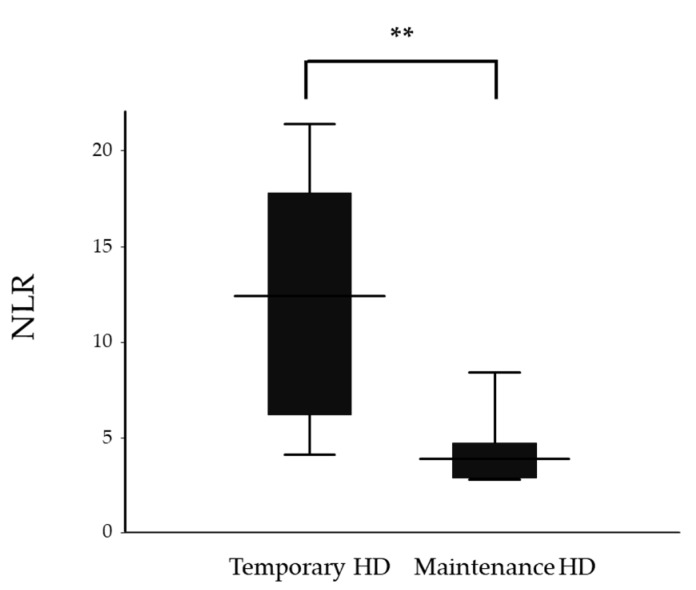
Neutrophil-to-lymphocyte ratios (NLR) of the temporary and maintenance hemodialysis (HD) groups. The top and the bottom of the boxes are the first and third quartile, respectively. The length of the box represents the interquartile range. The line through the middle of each box represents the median. The error bars show the minimum and maximum values (range). ** *p* < 0.01. NLR—neutrophil-to-lymphocyte ratio; HD—hemodialysis.

**Figure 5 jcm-09-01128-f005:**
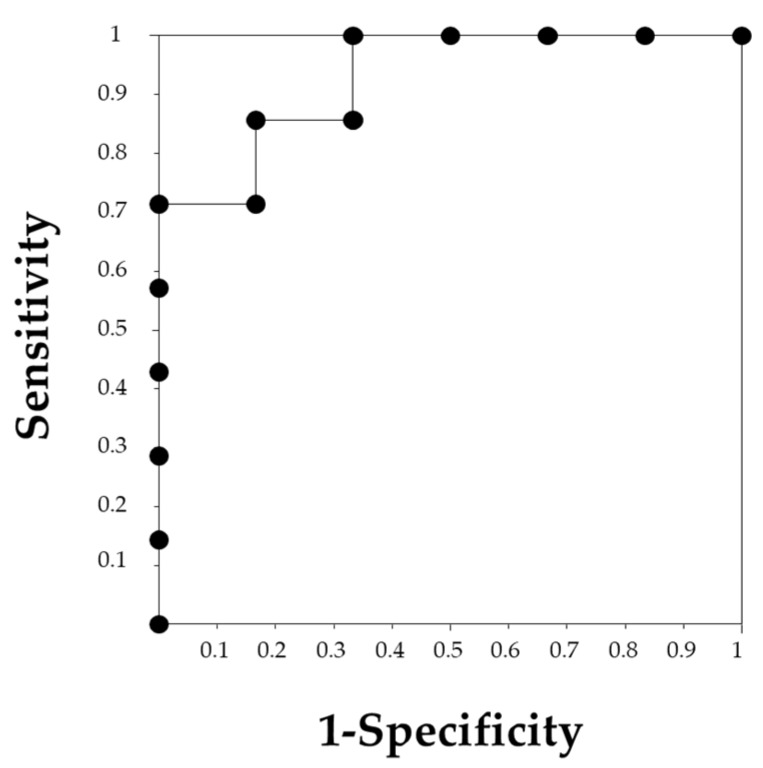
The receiver operating characteristic (ROC) curve for neutrophil-to-lymphocyte ratio (NLR) for recovery from renal failure. The NLR of 5.0 was determined to be a cutoff value with a sensitivity of 83.3% and a specificity of 85.7%, and the area under the curve was 0.929.

**Figure 6 jcm-09-01128-f006:**
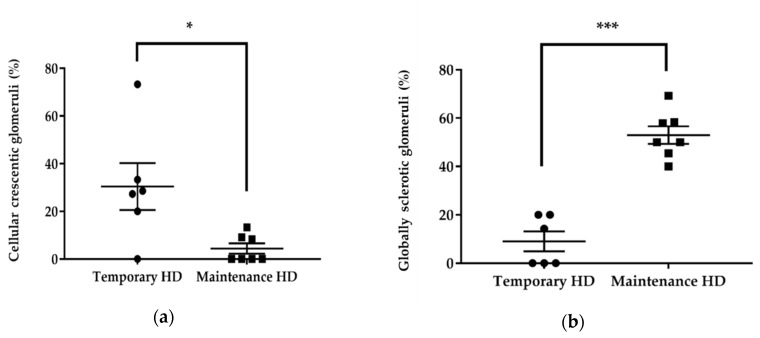
Quantification of histological findings in the temporary and maintenance hemodialysis (HD) groups. (**a**) Comparison of the percentage of glomeruli with cellular crescent between the temporary and maintenance HD groups. (**b**) Comparison of the percentage of globally sclerotic glomeruli between the temporary and maintenance HD groups. Bars indicate mean ± SEM. *, *p* < 0.05; *** *p* < 0.001. HD—hemodialysis.

**Figure 7 jcm-09-01128-f007:**
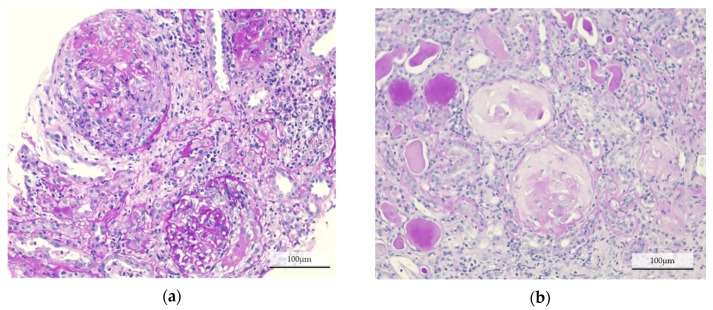
Histological findings in each group. Representative images of Periodic acid-Schiff staining on paraffin-embedded kidney sections, from patients in (**a**) the temporary hemodialysis (HD) group, and (**b**) the maintenance HD group. Cellular crescentic glomeruli were dominant in the temporary HD group, whereas most of the glomeruli were globally sclerotic in the maintenance HD group.

**Table 1 jcm-09-01128-t001:** Patient’s characteristics between the pre-dialysis and maintenance hemodialysis (HD) groups.

	Pre-dialysis (*n* = 37)	Maintenance HD (*n* = 7)	*p* Value
Sex (Male/Female)	22/15	5/2	0.132
Age (years)	71.4 ± 11.6	65.7 ± 7.8	0.222
Classifications of RPGN			
Immune complex CGN	33 (89.2%)	1 (14.3%)	
Pauci-immune CGN	4 (10.8%)	2 (28.6%)	
Anti-GBM CGN	0 (0%)	4 (57.1%)	
Immunosuppressive therapy			
Pulse corticosteroids	29/8	6/1	0.557
Cyclophosphamide	13/24	0/7	0.069
Plasma exchange	0/37	3/4	0.003
White blood cell count (10^3^/μL)	10.2 (4.9–23.5)	6.2 (4.7–12.4)	0.037
Neutrophil count (10^3^/μL)	8.5 (3.6–22.1)	4.6 (3.3–8.4)	0.012
Lymphocyte count (10^3^/μL)	1.17 ± 0.59	1.31 ± 0.43	0.550
Platelet count (10^3^/μL)	327 (98–808)	189 (117–269)	0.015
Creatinine (mg/dL)	2.80 ± 2.01	8.74± 1.80	<0.001
eGFR (mL/min/1.73 m^2^)	27.3 ± 21.2	5.4 ± 2.0	<0.001
CRP (mg/dL)	5.0 (0–24.8)	4.0 (0.4–26.9)	0.987
ESR (mm/h)	99 (10–140)	111 (62–134)	0.771
MPO-ANCA (U/mL)	166 (0–860)	0 (0–2440)	0.109
PR3-ANCA (U/mL)	0 (0–35.8)	0 (0–0)	0.308
anti-GBM antibody (U/mL)	0 (0–0)	42.3 (0–858.0)	<0.001
NLR	8.2 (2.0–32.0)	3.9 (2.8–8.4)	0.019
PLR	265.7 (82.9–2255.0)	126.0 (107.1–269.0)	0.008

Data are presented as the mean ± standard deviation, median (range), or number (%). HD—hemodialysis; RPGN—rapidly progressing glomerulonephritis; CGN—crescentic glomerulonephritis; eGFR—estimated glomerular filtration rate; CRP—C-reactive protein; ESR—erythrocyte sedimentation rate; MPO—myeloperoxidase; ANCA—anti-neutrophil cytoplasmic antibody; PR3—proteinase 3; GBM—glomerular basement membrane; NLR—neutrophil-to-lymphocyte ratio; PLR—platelet-to-lymphocyte ratio.

**Table 2 jcm-09-01128-t002:** Patient’s characteristics between temporary the HD and maintenance HD groups.

	Temporary HD (*n* = 6)	Maintenance HD (*n* = 7)	*p* Value
Sex (Male/Female)	1/5	5/2	0.048
Age (years)	72.7 ± 18.4	65.7 ± 7.8	0.381
Classifications of RPGN			
Immune complex CGN	0 (0%)	1 (14.3%)	
Pauci-immune CGN	6 (100%)	2 (28.6%)	
Anti-GBM CGN	0 (0%)	4 (57.1%)	
Immunosuppressive therapy			
Pulse corticosteroids	6/0	6/1	0.538
Cyclophosphamide	1/5	0/7	0.462
Plasma exchange	6/0	3/4	0.049
White blood cell count (10^3^/μL)	12.6 ± 5.1	7.6 ± 2.9	0.048
Neutrophil count (10^3^/μL)	11.0 ± 5.0	5.3 ± 2.1	0.018
Lymphocyte count (10^3^/μL)	1.10 ± 0.58	1.31 ± 0.43	0.470
Platelet count (10^3^/μL)	289.7 ± 112.7	191.4 ± 55.6	0.066
CRP (mg/dL)	11.2 ± 7.0	8.1 ± 10.3	0.554
ESR (mm/h)	104 ± 32	103 ± 27	0.985
MPO-ANCA (U/mL)	160 (17.0–469.0)	0 (0–2440)	0.138
anti-GBM antibody (U/mL)	0 (0–0)	42.3 (0–858.0)	0.065
NLR	12.4 (4.1–21.4)	3.9 (2.8–8.4)	0.008
PLR	341.7 ± 217.7	156.1 ± 62.6	0.053

Data are presented as the mean ± standard deviation, median (range), or number (%). HD—hemodialysis; RPGN—rapidly progressing glomerulonephritis; CGN—crescentic glomerulonephritis; eGFR—estimated glomerular filtration rate; CRP—C-reactive protein; ESR—erythrocyte sedimentation rate; MPO—myeloperoxidase; ANCA—anti-neutrophil cytoplasmic antibody; PR3—proteinase 3; GBM—glomerular basement membrane; NLR—neutrophil-to-lymphocyte ratio; PLR—platelet-to-lymphocyte ratio.

**Table 3 jcm-09-01128-t003:** Histological changes in the temporary hemodialysis (HD) and maintenance HD groups.

	Temporary HD (*n* = 6)	Maintenance HD (*n* = 7)	*p* Value
Cellular crescent (%)	30.4 ± 24.1	4.4 ± 5.7	0.022
Fibrocellular crescent (%)	11.9 ± 14.1	10.2 ± 9.4	0.945
Fibrous crescent (%)	19.0 ± 22.9	20.1 ± 16.3	0.921
Global sclerosis (%)	9.0 ± 10.1	53.0 ± 9.7	<0.001

Data are presented as the mean ± standard deviation. HD—hemodialysis.

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
