# Peer review of "Prognostic Value of Neutrophil-To-Lymphocyte Ratio and Platelet-To-Lymphocyte Ratio for Renal Outcomes in Patients with Rapidly Progressive Glomerulonephritis"

_jcm, 2020, doi:10.3390/jcm9041128_

Round 1
Reviewer 1 Report
The authors did a retrospective single center study to look at the predictive value of NLR and PLR in RPGN patients. The study is fairly well done. Few issues are as follows:
- Moderate English Language edition required.
- The Table 1 and Table 2 are important tables from where authors derived their conclusions. However, given that the range of data is so large, just showing median is not acceptable. Please show a five-number summary using a box-plot to show your data for both Table and Table 2, mainly for the NLR and PLR data.
- In Table 1, the pre-dialysis group didn't have Anti-GBM disease. Could this have caused a bias in your NLR and PLR result?
- How could you explain the statement that" pathologist was blinded for the clinical data" when you are doing a retrospective study of 10 years? Do you need to modify that statement?
Author Response
Comments and Suggestions for Authors
The authors did a retrospective single center study to look at the predictive value of NLR and PLR in RPGN patients. The study is fairly well done. Few issues are as follows:
Thank you very much for reviewing the manuscript together with a constructive comment. Our responses to the comments are below:
- Moderate English Language edition required.
Response: The manuscript underwent English editing.
- The Table 1 and Table 2 are important tables from where authors derived their conclusions. However, given that the range of data is so large, just showing median is not acceptable. Please show a five-number summary using a box-plot to show your data for both Table and Table 2, mainly for the NLR and PLR data.
Response: We have added the box-plot figure for NLR and PLR (Figure 2,4)
- In Table 1, the pre-dialysis group didn't have Anti-GBM disease. Could this have caused a bias in your NLR and PLR result?
Response: We have defined the study subjects as “RPGN”, in which anti-GBM disease is included. Since 4 cases of anti-GBM disease required maintenance HD, this might affect the results. However, we observed a significant difference in NLR as well when patients with anti-GBM disease were eliminated (data not shown). We have incorporated this point into the “limitation”. We have added “Since all four patients with anti-GBM disease required maintenance HD, this may affect the result of our study. However, we observed a significant difference in NLR between the temporary and maintenance HD group when these patients were eliminated.”
- How could you explain the statement that" pathologist was blinded for the clinical data" when you are doing a retrospective study of 10 years? Do you need to modify that statement?
Response: Sorry for the confusing expression. We have modified the statement to “…who was independent of the acquisition and analysis of the clinical information”.
Reviewer 2 Report
- The small sample size in this study, as well as its retrospective nature, without pre-set outcomes, does not allow outright conclusions. The value of the results cannot be completely denied but should be acknowledged throughout the text as only hypothetical, with avoidance of categorical expressions.
- In order to maintain scientific objectivity, studies not showing positive results for NLR and PLR should also be presented and interpreted within the general context in the Introduction and Discussion sections.
- The limitations of the study are serious. Even though the small sample size and the retrospective nature of the study are already shortly mentioned in the text, the adequate size of the credibility they cast on the scientific results should be properly acknowledged. I would also mention here the large variance seen in the confidence intervals for the NLR and PLR values in pre-dialysis patients, which also limits the strength of evidence.
- The expression “during the course”, seen at page 4, line 106 and elsewhere in the text is unclear or clumsy in the context of the surrounding phrases and might warrant its replacement with more appropriate terms.
- The quality of the English language is altogether good. However, some isolated phrases or expressions contain clumsy English constructs or syntactic errors (e.g., disagreements between noun and verb, wrong verb mode, wrong singular/plural noun form, etc.), so rethinking them might prove beneficial. I quote some of the problematic constructions below:
“than in patients required maintenance hemodialysis” (page 1, line 21)
“We propose that NLR and PLR can be predictive markers” (page 1, line 26)
“44 patients were included in the analyses” (page 2, lines 57-58)
“Renal outcome after 1 year from diagnosis” (page 2, line 67)
“the 13 patient” (page 4, line 106)
“and confirmed no patients had history” (page 6, line 153)
“higher NLR and PLR indicates” (page 6, line 158)
“lower NLR and PLR suggests” (page 6, line 159)
Author Response
Comments and Suggestions for Authors
- The small sample size in this study, as well as its retrospective nature, without pre-set outcomes, does not allow outright conclusions. The value of the results cannot be completely denied but should be acknowledged throughout the text as only hypothetical, with avoidance of categorical expressions.
Response: We have modified the misleading expressions throughout the manuscript. Besides, we have modified the sentences in the “limitation” as follows; “ the results of the present study should be carefully interpreted”.
- In order to maintain scientific objectivity, studies not showing positive results for NLR and PLR should also be presented and interpreted within the general context in the Introduction and Discussion sections.
Response: Although we could find only a few reports describing negative results, we have modified and added some references. We have added “On the other hand, several studies have mentioned that the application of NLR and PLR is limited. It has been demonstrated that NLR is a good predictor of the relapse rate but not death in patients with AAV [22]. PLR is also able to predict the disease activity but cannot predict relapse in AAV patients [20].” in the “Discussion” section.
- The limitations of the study are serious. Even though the small sample size and the retrospective nature of the study are already shortly mentioned in the text, the adequate size of the credibility they cast on the scientific results should be properly acknowledged. I would also mention here the large variance seen in the confidence intervals for the NLR and PLR values in pre-dialysis patients, which also limits the strength of evidence.
Response: We understand the point. The sample size and the retrospective design are the limitations of this study. We have added the study cohort as an information in the “Materials and Methods” section. We have mentioned about the need for careful interpretation and the perspectives of this theme. As the reviewer pointed out, the variations in NLR and PLR were considered to be the limitations in the present study. We have added “In addition to the variations in NRL and PLR, the study is a retrospective study with a small number of the subject.” in the limitation.
- The expression “during the course”, seen at page 4, line 106 and elsewhere in the text is unclear or clumsy in the context of the surrounding phrases and might warrant its replacement with more appropriate terms.
Response: We have removed the expression in the text.
- The quality of the English language is altogether good. However, some isolated phrases or expressions contain clumsy English constructs or syntactic errors (e.g., disagreements between noun and verb, wrong verb mode, wrong singular/plural noun form, etc.), so rethinking them might prove beneficial. I quote some of the problematic constructions below:
“than in patients required maintenance hemodialysis” (page 1, line 21)
“We propose that NLR and PLR can be predictive markers” (page 1, line 26)
“44 patients were included in the analyses” (page 2, lines 57-58)
“Renal outcome after 1 year from diagnosis” (page 2, line 67)
“the 13 patient” (page 4, line 106)
“and confirmed no patients had history” (page 6, line 153)
“higher NLR and PLR indicates” (page 6, line 158)
“lower NLR and PLR suggests” (page 6, line 159)
Response: Thank you for pointing these out. We have modified the English.
Submission Date
26 March 2020
Date of this review
02 Apr 2020 23:12:14
Reviewer 3 Report
The authors examined the prognostic value of neutrophil-to-lymphocyte ratio and platelet-to-lymphocyte ratio for renal outcome in patients with rapidly progressive glomerulonephritis.
The idea of this study is good. The manuscript is well written overall.
Comments:
The definition and classification of rapidly progressing glomerulonephritis should be described in more detail in the introduction.
The description of patients is insufficient. Inclusion and exclusion criteria as well as biopsy indications should be described in more detail.
It would be of interest to specify the ethnic features of the population.
The therapy of patients should be described more precisely.
A more detailed description of statistical methods is required (especially ROC analysis).
Multivariarte regression analysis should be performed. It is also important to note that the authors should consider all variables that may affect the outcome.
Author Response
Comments and Suggestions for Authors
The authors examined the prognostic value of neutrophil-to-lymphocyte ratio and platelet-to-lymphocyte ratio for renal outcome in patients with rapidly progressive glomerulonephritis.
The idea of this study is good. The manuscript is well written overall.
Comments:
We appreciate the reviewer’s effort to review the manuscript, and thank the reviewer for the positive appreciation of the work.
The definition and classification of rapidly progressing glomerulonephritis should be described in more detail in the introduction.
Response: We have added those in the “Introduction” section as follows; “The diagnosis of RPGN is made when renal dysfunction occurs within a short period of time and is complicated with proteinuria or hematuria [2]. The etiology of RPGN is divided into three classifications; immune complex crescentic glomerulonephritis, pauci-immune crescentic glomerulonephritis, and anti-glomerular basement membrane (GBM) crescentic glomerulonephritis.”
The description of patients is insufficient. Inclusion and exclusion criteria as well as biopsy indications should be described in more detail.
Response: Indications of the renal biopsy was based on the guideline. This was incorporated in the “Study population” section as follows; “Renal biopsies were performed according to the indications of the guidelines from the Japanese Society of Nephrology [15]; persisting hematuria and/or proteinuria, proteinuria more than 0.5 g/day, rapid decline in renal function, and gross hematuria.”
Furthermore, we have added “Figure 1” describing the study population.
It would be of interest to specify the ethnic features of the population.
Response: We agree, however all the patients enrolled in this study were Japanese. We have added the following description in the “results” section; “All the patients enrolled in this study were ethnically homogenous.”
The therapy of patients should be described more precisely.
Response: The therapy was determined based on the guideline from the Japanese Society of Nephrology [15]. The detailed therapies provided to the patients were summarized and added in the Tables. We have modified the tables according to the comment.
A more detailed description of statistical methods is required (especially ROC analysis).
Response: we have modified the “statistical analyses” section. We have added “The Kolmogorov-Smirnov test was used to assess normal distribution.” and “The optimal cutoff point was determined by minimizing the square of the distance between the point (sensitivity of 1, 1-specificity of 0) and any point on ROC curve.”
Multivariarte regression analysis should be performed. It is also important to note that the authors should consider all variables that may affect the outcome.
Response: We additionally performed the multivariate regression analysis to investigate the influencing factor for renal outcome in 44 patients. In addition to renal function, which is a known variable for renal outcome, there was also a trend toward significance for NLR.
Since renal function is doubtlessly a strong predictor for renal outcome, we have analyzed the usefulness of NLR and PLR in patients required HD (temporary HD and maintenance HD) in order to eliminate the contribution of renal function. This was mentioned in the “Results” section. We have added “…the multivariate analysis revealed that renal function was the strongest influencing factor, we investigated the predictive abilities of NLR and PLR in patients required HD”. in the “Discussion section.
Round 2
Reviewer 1 Report
Much better appearing now.
Reviewer 2 Report
The authors carefully responded to each comment, allowing the overall quality of the manuscript to improve considerably.
I think the current form of this manuscript can be considered for publication by the journal.
Reviewer 3 Report
The authors have addressed the comments of the reviewer.